# GmTCP40 Promotes Soybean Flowering under Long-Day Conditions by Binding to the *GmAP1a* Promoter and Upregulating Its Expression

**DOI:** 10.3390/biom14040465

**Published:** 2024-04-10

**Authors:** Lixin Zhang, Peiguo Wang, Miao Wang, Xin Xu, Hongchang Jia, Tingting Wu, Shan Yuan, Bingjun Jiang, Shi Sun, Tianfu Han, Liwei Wang, Fulu Chen

**Affiliations:** Ministry of Agriculture and Rural Affairs Key Laboratory of Soybean Biology (Beijing), Institute of Crop Sciences, Chinese Academy of Agricultural Sciences, 12 Zhongguancun South Street, Beijing 100081, China; zhanglixin_1994@163.com (L.Z.); wpg15101224039@163.com (P.W.); wangmiao200161@163.com (M.W.); xuxin_caas@163.com (X.X.); jiahongchangheihe@163.com (H.J.); wutingting@caas.cn (T.W.); yuanshan@caas.cn (S.Y.); jiangbingjun@caas.cn (B.J.); sunshi@caas.cn (S.S.); hantianfu@caas.cn (T.H.)

**Keywords:** soybean, flowering time, *GmTCP40*, *GmAP1a*, photoperiod

## Abstract

Soybean [*Glycine max* (L.) Merr.] is a short-day (SD) plant that is sensitive to photoperiod, which influences flowering, maturity, and even adaptation. TEOSINTE-BRANCHED1/CYCLOIDEA/PROLIFERATING CELL FACTOR (TCP) transcription factors have been shown to regulate photoperiodic flowering. However, the roles of TCPs in SD plants such as soybean, rice, and maize remain largely unknown. In this study, we cloned the *GmTCP40* gene from soybean and investigated its expression pattern and function. Compared with wild-type (WT) plants, *GmTCP40*-overexpression plants flowered earlier under long-day (LD) conditions but not under SD conditions. Consistent with this, the overexpression lines showed upregulation of the flowering-related genes *GmFT2a*, *GmFT2b*, *GmFT5a*, *GmFT6*, *GmAP1a*, *GmAP1b*, *GmAP1c*, *GmSOC1a*, *GmSOC1b*, *GmFULa*, and *GmAG* under LD conditions. Further investigation revealed that GmTCP40 binds to the *GmAP1a* promoter and promotes its expression. Analysis of the *GmTCP40* haplotypes and phenotypes of soybean accessions demonstrated that one *GmTCP40* haplotype (Hap6) may contribute to delayed flowering at low latitudes. Taken together, our findings provide preliminary insights into the regulation of flowering time by *GmTCP40* while laying a foundation for future research on other members of the *GmTCP* family and for efforts to enhance soybean adaptability.

## 1. Introduction

Soybean [*Glycine max* (L.) Merr.] is a short-day (SD) plant that is sensitive to photoperiod. Originally temperate plants, soybean plants have been grown over a wide range of latitudes worldwide, from 53° N to 35° S [1]. Importantly, the photoperiod sensitivity of a soybean variety determines where it can be planted. Because photoperiod sensitivity determines the timing of flowering and maturity, if a soybean variety is planted in a region with inappropriate day length, the plants will fail to mature. Therefore, it is important to elucidate the genetic networks of photoperiodic flowering to improve the adaptability of soybean to different regions.

A basic model for the regulation of flowering time in soybean has been established in which *E3*/*E4*-*E1*-*GmFLOWERING LOCUS Ts* (*FTs*) control flowering under different photoperiods. *E1* is an inhibitor of the soybean flowering pathway, regulating the expression of *GmFTs* [2,3,4], and E3 (phytochrome A3 [PHYA3]) and E4 (PHYA2) directly bind to and stabilize E1 and its homologs [5]. FT proteins integrate signals and are transported from the leaves to the shoot apex to regulate flowering [6]. In soybean, *GmFTs* regulate flowering by indirectly influencing the expression of floral meristem identity genes such as *GmAPETALA1a* (*AP1a*), *GmAP1b*, *GmSUPPRESSOR OF OVEREXPRESSOR OF CONSTANS 1a* (*SOC1a*), and *GmSOC1b*.

The plant-specific TEOSINTE-BRANCHED1/CYCLOIDEA/PROLIFERATING CELL FACTOR (TCP) transcription factor, which is a key regulator of plant growth and development [7], also regulates photoperiodic flowering by directly or indirectly regulating the expression of floral integrators such as *FT*, *CONSTANS* (*CO*), *SOC1*, and *AP1*. The TCP transcription factor family was named based on the first characterized members, namely TEOSINTE BRANCHED1 (TB1) in maize (*Zea mays*), CYCLOIDEA (CYC) in snapdragon (*Antirrhinum majus*), and PROLIFERATING CELL FACTOR 1 (PCF1) and PCF2 in rice (*Oryza sativa*), which each contain a noncanonical basic helix–loop–helix motif referred to as the TCP domain [8,9,10]. The TCP family consists of two classes (Class I and Class II) that are distinguishable by specific amino acids in the TCP domain [11,12]. The Class II TCP members are divided into two subclasses: CINCINNATA (CIN) and CYC/TB1 [12,13].

In *Arabidopsis*, the Class I member TCP7 interacts with nuclear factor Y proteins and CO to promote flowering by regulating the expression of *SOC1* [14]; TCP15 promotes flowering by regulating *SOC1* expression [15]; and *TCP8* and *TCP23* contribute to flowering time determination by regulating *FT* and *SOC1* expression [16]. The Class II TCP members TCP2, TCP3, TCP4, TCP10, and TCP24 are miR319-target genes that promote flowering by positively regulating *CO* expression [17]. TCP4 physically interacts with the flowering promoter GIGANTEA (GI) and promotes *CO* expression in a GI-dependent manner [18]. TCP5, TCP13, and TCP17 are integrated into the FT-FD complex and regulate flowering by directly binding to the promoter of *AP1* [19]. TCP20 and TCP22 interact with LIGHT-REGULATED WD1 (*LWD1*) and activate the expression of the circadian clock pathway gene *CIRCADIAN CLOCK ASSOCIATED1* (*CCA1*) to inhibit flowering [20]. TCPs have also been shown to play roles in flowering time regulation in soybean. *GmTCP13*, a member of the CIN subclass, has been identified as a candidate gene for *QNE1* (QTL near E1) and functions as a positive regulator of flowering by enhancing the expression of *GmFT2a* and *GmFT5a* [21]. In addition, GmMRF2 promotes flowering by interacting with another member of the CIN subclass, GmTCP15, to induce the expression of *GmSOC1b* [22]. However, limited research has been conducted on the regulatory roles of *GmTCPs* in flowering, particularly of Class I *GmTCPs*.

In the present study, we cloned the *GmTCP40* gene, analyzed its expression pattern, and identified its function in regulating flowering time. We also analyzed its flowering time regulatory mechanism and performed haplotype analysis. We found that GmTCP40 induces flowering by upregulating the expression of the downstream flowering-related gene *GmAP1a* under long-day (LD) conditions but not under SD conditions. Association analysis of haplotypes with flowering time across 175 resequenced soybean varieties revealed that one *GmTCP40* haplotype, *GmTCP40-*Hap6, may contribute to later flowering at low latitudes. Taken together, our results indicate that *GmTCP40* plays an important role in soybean flowering and may contribute to soybean adaptability.

## 2. Materials and Methods

### 2.1. Phylogenetic Analyses

The sequences of the TCP proteins from *Arabidopsis thaliana*, rice (*Oryza sativa*), maize (*Zea mays*), and soybean for phylogenetic analyses were downloaded from the Phytozome13 database (https://phytozome-next.jgi.doe.gov/, accessed on 8 July 2021). Phylogenetic trees for the TCP protein family were constructed using the neighbor-joining method, with a bootstrap value of 1000, in MEGA 7 software (https://www.megasoftware.net/, accessed on 15 July 2021) [23]. Then, the phylogenetic tree was visualized using the online tool Evolview (https://www.evolgenius.info/evolview/, accessed on 20 October 2023).

### 2.2. Plant Materials and Growth Conditions

The soybean varieties “Zigongdongdongdou (ZGDD)” and “Heihe27 (HH27)” were planted in controlled culture rooms at 26 °C under SD (12 h light/12 h dark) and LD (16 h light/8 h dark) conditions for analysis of the expression pattern of *GmTCP40* (*Glyma16G004300*). ZGDD and HH27 have been used as model genotypes in the study of photoperiod responses. The photoperiod-sensitive variety ZGDD belongs to maturity group (MG) VIII, with genotype *E1*/*E2*/*E3*/*E4*. The photoperiod-insensitive variety HH27 belongs to MG 0, with genotype *e1-as*/*e2-ns*/*e3-tr*/*E4* [24]. The soybean variety “Jack” was used to examine tissue-specific expression profiles of *GmTCP40*. Cotyledons and shoot apical meristems (SAMs) were collected 15 days after the emergence of soybean cotyledons from the soil surface (VE) [25]. After the plants had flowered, different organs (roots, leaves, stems, and flowers) were sampled for RNA extraction.

The soybean variety “Jack” was used for plant transformation. All the transgenic and wild-type (WT) plants were grown under SD and LD conditions for phenotypic measurements. There were five pots for each of the three transgenic lines and WT plants, with each pot containing four seeds. The planting substrate consisted of a mixture of soil and vermiculite in a ratio of 1:2. We collected leaves from the transgenic and WT plants for RNA extraction.

A diverse panel of 175 soybean varieties, namely 88 varieties from Northeast China, 49 varieties from Huang-Huai-Hai, and 38 varieties from South China, was used for *GmTCP40* haplotype analysis (Appendix A). The panel was planted at seven different locations during 2016 and 2017: Sanya (18°18′ N, 112°39′ E) in 2016, Xiangtan (27°40′ N, 112°39′ E) in 2016, Jining (35°26′ N, 116°35′ E) in 2016, Xinxiang (35°08′ N, 113°45′ E) in 2016 and 2017, Beijing (40°13′ N, 116°33′ E) in 2016 and 2017, Changchun (43°50′ N, 124°82′ E) in 2017, and Heihe (50°24′ N, 127°49′ E) in 2017. These nine environments were named SY2016, JN2016, XX2016, BJ2016, XT2016, XX2017, BJ2017, CC2017, and HH2017, respectively (Appendix A). All the materials were arranged in randomized complete blocks with two replications. The flowering time across these nine environments for the 175 soybean varieties was recorded as the days from VE to the beginning of blooming (R1), defined as the appearance of the first flower at any node on the main stem [25]. The average flowering time of two replicates was used for analysis.

### 2.3. RNA Extraction and qPCR

Total RNA was extracted using the RNA Easy Fast Plant Tissue Kit (TIANGEN Biotech [Beijing] Co., Ltd., China, Cat#DP452). cDNAs were synthesized with the FastKing RT kit (with gDNA) (TIANGEN Biotech [Beijing] Co., Ltd., Beijing, China, Cat#KR116). Quantitative PCR (qPCR) was performed using Taq Pro Universal SYBR qPCR Master Mix (Vazyme Biotech Co., Ltd., Nanjing, China, Cat#Q712-02). Three replicates were performed for each sample. The 2^−ΔΔCt^ method was used for the calculation of relative expression levels [26]. The primers used for qPCR are listed in Appendix A.

### 2.4. Plasmid Construction and Plant Transformation

The coding sequence (CDS) of *GmTCP40* was cloned from “Jack” to construct the overexpression vector. The amplified cDNA was inserted into an *Xba*I-digested pTF101 vector [27] using the ClonExpress^®^ Ultra One Step Cloning Kit (Vazyme Biotech Co., Ltd., Nanjing, China, Cat#C115). Expression of the GmTCP40-GFP fusion protein was driven by the CaMV 35S promoter (Appendix A). The recombinant vector pTF101-*GmTCP40* was subsequently introduced into *Agrobacterium tumefaciens* strain EHA101 (Beijing Zoman Biotech Co., Ltd., Beijing, China, Cat#ZC1405) as described in Supplementary Methods. pTF101-*GmTCP40* was subsequently transformed into the soybean variety “Jack” according to a previous protocol [28]. Transgenic plants were identified using Liberty Link strips with PAT proteins (Aojin Biotech [Tianjin] Co., Ltd., Tianjin, China, Cat#AG-002-SLF) and PCR. The primers used for plasmid construction are listed in Appendix A.

### 2.5. Subcellular Localization of GmTCP40

The *Agrobacterium tumefaciens* strain GV3101 carrying pTF101-*GmTCP40* was injected into the leaves of *Nicotiana benthamiana* as described by Kubota et al. [18]. The fluorescence signals of GmTCP40-GFP were observed and imaged using an FV3000 confocal microscope (Olympus Corp., Shinjuku-ku, Japan).

### 2.6. Phenotyping and Statistical Analysis

The flowering times of the transgenic plants and WT plants were recorded as the number of days from VE to R1 [25]. All the statistical analyses (Student’s *t*-tests) were performed using GraphPad 8.

### 2.7. Haplotype Analysis of GmTCP40

To investigate the natural variation in *GmTCP40*, we analyzed the resequencing data of 175 varieties which were downloaded from the National Center for Biotechnology Information (NCBI) database under the Short Read Archive (SRA) accession numbers SRP062560 and PRJNA589345 [29]. Using Tassel 5 software, we changed the format to intuitively observe the changes in nucleotides compared with the reference genome Wm82.a2.v1 (https://phytozome-next.jgi.doe.gov/info/Gmax_Wm82_a2_v1, accessed on 13 September 2023). We conducted an association analysis of *GmTCP40* haplotypes with flowering time using SPSS 21.0 via Duncan’s multiple range test using (* *p* < 0.05). GraphPad Prism 8 and Excel 2019 were used for visual mapping.

### 2.8. Yeast One-Hybrid System

The EGY48-LacZ system was employed for yeast one-hybrid (Y1H) assays (Clontech Laboratories, Inc., Mountain View, CA, USA). For the construction of the pB42AD-*GmTCP40* vector, the pB42AD vector was digested with the *EcoR*I and *Xho*I restriction enzymes, followed by seamless ligation of the purified *GmTCP40* CDS fragment into the linearized pB42AD vector using the ClonExpress^®^ Ultra One Step Cloning Kit (Vazyme Biotech Co., Ltd., Nanjing, China, Cat#C115). A 3000 bp fragment of the *GmAP1a* promoter was amplified from the soybean variety “Jack”. The amplified fragment was subcloned into the pLacZ2u vector digested with the *Kpn*I and *Xho*I restriction enzymes using the ClonExpress^®^ Ultra One Step Cloning Kit (Vazyme Biotech Co., Ltd., Nanjing, China, Cat#C115). Three fragments (967 bp, 2955 bp, and 5541 bp) of the *GmSOC1a* promoter were amplified and inserted into different pLacZ2u vectors (Appendix A). Meanwhile, 1243 bp, 2906 bp, and 4943 bp fragments of the *GmSOC1b* promoter were cloned into different pLacZ2u vectors (Appendix A). Appropriate pairs of constructs were subsequently transformed into the yeast strain EGY48, as described in the Supplementary Methods. The yeast clones were grown on SD/-Trp/-Ura media (Beijing Coolabi Technology Co., Ltd., Beijing, China, Cat#PM2262), which was selected for yeast clones containing the desired plasmids, at 30 °C for 3 days. Then, the yeast clones were spotted onto SD media (lacking Trp and Ura) supplemented with X-gal, Galactose (Gal), and Raffinose (Raf) to detect interactions. The primers used for plasmid construction are listed in Appendix A.

### 2.9. Transient Luciferase Reporter Assays

To generate effector constructs, the CDS of *GmTCP40* (with *Kpn*I and *Pst*I restriction sites on the primers to amplify the CDS) was inserted into the pGreenII 62-SK vector using the ClonExpress^®^ Ultra One Step Cloning Kit (Vazyme Biotech Co., Ltd., Nanjing, China, Cat#C115). The amplified promoter of *GmAP1a (*with *Kpn*I and *Pst*I restriction sites on the primers to amplify the promoter) was subcloned into the pGreenII 0800-LUC vector using the ClonExpress^®^ Ultra One Step Cloning Kit (Vazyme Biotech Co., Ltd., Nanjing, China, Cat#C115). The Renilla luciferase (REN) was used as an internal control. The relevant constructs were introduced into *Agrobacterium* strain EHA105 (pSoup) (Beijing Zoman Biotech Co., Ltd., Beijing, China, Cat#ZC1048) and subsequently co-infiltrated into tobacco leaves, as described by Kubota et al. [18]. Two days after infiltration, the injected leaves were sprayed with 2.5 mM beetle luciferin potassium salt (Promega Corp., Madison, USA, Cat#E1601) and then incubated in darkness for 5 min before being transferred to a Tannon5200 (Shanghai, China) for capturing the luciferase signals. Then, we collected leaves and performed extraction followed by the addition of substrates for REN and LUC separately to evaluate their activities with the Dual-Luciferase Reporter Gene Assay Kit (Yeasen Biotech Co., Ltd., Shanghai, China, Cat#11402ES60). The primers used for plasmid construction are listed in Appendix A.

## 3. Results

### 3.1. Cloning of GmTCP40 and Determination of Its Expression Patterns

A phylogenetic tree of the TCP family was constructed using TCP proteins from *Arabidopsis*, rice, maize, and soybean (Appendix A). In soybean, 54 TCPs were identified and classified into two classes: Class I (26 GmTCPs) and Class II (28 GmTCPs) [6]. Class II was divided into two subclasses: CIN (19 GmTCPs) and CYC/TB (9 GmTCPs). The flowering regulators GmTCP13 and GmTCP15 were assigned to the CIN subclass, and their *Arabidopsis* homologs are associated with flowering time regulation (Appendix A). The regulatory roles of numerous Class I genes such as *AtTCP7*, *AtTCP15*, *AtTCP22*, and *AtTCP23* in photoperiodic flowering have been extensively investigated in *Arabidopsis* [14,15,16,20,30]. However, the precise regulatory roles of these TCPs in soybean remain to be fully elucidated. Here, we studied the soybean gene *GmTCP40*, a gene closely related to *Arabidopsis* Class I TCPs (Appendix A).

The expression patterns of *GmTCP40* were initially investigated in the early-maturity soybean variety “HH27” and late-maturity soybean variety “ZGDD” under SD and LD conditions. HH27 flowered at 28.1 ± 1.7 days under SD condition and 27.6 ± 1.8 days under LD condition [31], while ZGDD flowered at 32.0 ± 2.5 days under SD conditions and maintained vegetative growth under LD conditions [31]. In both the HH27 and ZGDD plants, the expression of *GmTCP40* peaked at 12 h after dawn under both SD and LD conditions (Figure 1A,B). The expression level of *GmTCP40* was higher in HH27 than in ZGDD under SD and LD conditions.

We subsequently examined the expression levels of *GmTCP40* in various organs of the “Jack” variety under SD conditions. Notably, *GmTCP40* was expressed predominantly in the cotyledons, roots, and SAM (Figure 1C), while lower expression levels were detected in the stem, leaf, and flower tissues.

To investigate the subcellular localization of the GmTCP40 protein, we generated a construct expressing a GmTCP40-GFP fusion protein and introduced it into *N. benthamiana* leaves through infiltration. Fluorescence signals from GmTCP40-GFP were observed within the nuclei of epidermal cells in *N. benthamiana* leaves (Figure 1D).

### 3.2. Overexpression of GmTCP40 Promotes Soybean Flowering under LD Conditions

To further elucidate the regulatory role of *GmTCP40* in flowering time, we cloned the CDS of *GmTCP40* from the soybean variety “Jack” and generated transgenic plants overexpressing *GmTCP40* under the control of the CaMV 35S promoter in the “Jack” variety. The transgenic plants were confirmed by the PAT protein test and PCR and qPCR analyses. Three transgenic lines were selected for phenotypic characterization. Under SD conditions, the three GmTCP40-overexpressing (OE) lines, OE-1 (23.33 days), OE-2 (23.00 days), and OE-3 (23.57 days), flowered slightly earlier than WT plants, which flowered at 24.29 days, but these differences were not significant (Figure 2A,C). However, under LD conditions, OE-1 (51.67 days), OE-2 (51.71 days), and OE-3 (50.64 days) plants flowered significantly earlier than the WT plants (55.08 days), a difference of 3.36–4.43 days (Figure 2B,D). These results indicate that *GmTCP40* acts as a positive regulator of flowering.

To elucidate the mechanisms by which *GmtTCP40* regulates flowering time, we analyzed the expression levels of *GmFTs* (*GmFT1a*, *GmFT2a*, *GmFT2b*, *GmFT3a*, *GmFT3b*, *GmFT5a*, *GmFT5b*, and *GmFT6*), which are integrative factors in the flowering pathway. In addition, we examined the expression profiles of several downstream genes (*GmAP1a*, *GmAP1b*, *GmAP1c*, *GmSOC1a, GmSOC1b*, *GmFRUITFULa* [*FULa*], and *GmAGAMOUS* [*AG*]) in the WT and *GmTCP40*-OE plants. Under LD conditions, the *GmTCP40*-OE lines exhibited significantly higher expression levels of *GmFT2a*, *GmFT2b*, *GmFT5a*, *GmFT6*, *GmAP1a*, *GmAP1b*, *GmAP1c*, *GmSOC1a*, *GmSOC1b*, *GmFULa*, and *GmAG* than the WT plants; however, the expression levels of *GmFT1a* and *GmFT3a* were significantly lower in the OE-*GmTCP40* plants than in the WT plants (Figure 3). There were no significant differences in the expression levels of *GmFT3b* or *GmFT5b* between the *GmTCP40*-OE and WT plants (Figure 3). Under SD conditions, no significant changes were observed in the expression of any of the genes in either the WT or *GmTCP40*-OE plants (Appendix A). These results suggest that *GmTCP40* promotes flowering by regulating the expression of downstream flowering-related genes.

### 3.3. GmTCP40 Binds to the GmAP1a Promoter

In *Arabidopsis*, TCP5, TCP13, and TCP17 have been shown to induce flowering by binding to the promoter of *AP1* [19], and TCP7 and TCP15 bind to the *SOC1* promoter to promote flowering [14,15]. In this study, we investigated whether GmTCP40 could bind to the promoters of *GmAP1a*, *GmSOC1a,* and *GmSOC1b*, which were upregulated in *GmTCP40*-OE plants. We cloned the promoters of the homologs of *SOC1* (*GmSOC1a* and *GmSOC1b*) and *AP1* (*GmAP1a*). Y1H assays showed that GmTCP40 bound to the promoter of *GmAP1a* but not to the promoter of *GmSOC1a* or *GmSOC1b* (Figure 4A,B and Appendix A), indicating that *GmAP1a* is a target gene of GmTCP40. Next, we performed transient luciferase reporter assays, which revealed that GmTCP40 activated transcription of the *GmAP1a* (Figure 4C–E).

### 3.4. Haplotype Analysis of GmTCP40 Reveals Its Roles in Flowering and Regional Adaptation

To evaluate the impact of natural variation in *GmTCP40* on soybean adaptation, we examined the genotypes of *GmTCP40* in 175 soybean varieties originating from diverse geographical regions across China (Appendix A). No frameshifts or amino acid substitutions were found in the CDS of *GmTCP40*. A total of 14 polymorphic loci, namely 11 single-nucleotide polymorphisms (SNPs) and three insertion-deletions (indels), were detected in the promoter region, and six haplotypes (Hap1–Hap6) were defined (Figure 5A). We analyzed the geographical distributions of the major haplotypes and their association with flowering time in nine environments across China: Sanya (18°18′ N, 112°39′ E) in 2016, Xiangtan (27°40′ N, 112°39′ E) in 2016, Jining (35°26′ N, 116°35′ E) in 2016, Xinxiang (35°08′ N, 113°45′ E) in 2016 and 2017, Beijing (40°13′ N, 116°33′E) in 2016 and 2017, Changchun (43°50′ N, 124°82′ E) in 2017, and Heihe (50°24′ N, 127°49′ E) in 2017 (Appendix A; Appendix A). *GmTCP40*-Hap3 was widely distributed across China and had flowering times similar to those of *GmTCP40*-Hap2 and *GmTCP40*-Hap4 (Figure 5B,C). *GmTCP40*-Hap6 exhibited a later flowering time in eight environments (Figure 5B), and the frequency of *GmTCP40*-Hap6 decreased with increasing latitude in China (Figure 5C). Analysis of the *cis*-elements in the *GmTCP40* promoter region revealed two Box-4 and CAAT-box elements located between positions 239,601 and 239,616 (Appendix A). Box-4 is involved in the light response and the transcriptional regulation of many plant genes specifically expressed in flower organs. The CAAT-box is located in the promoter and enhancer regions of *GmTCP40*. A large sequence deletion from bp 239,601 to 239,616 in *GmTCP40*-Hap6 accessions may account for late flowering. These results indicated that naturally occurring mutations in *GmTCP40* play an important role in soybean adaptation to a wide range of latitudes in China, and further characterization of *GmTCP40*-Hap6 might facilitate the genetic improvement of soybean.

## 4. Discussion

In *Arabidopsis*, *TCP* family members exhibit diverse functions in photoperiodic flowering regulation. Class I TCPs, such as *TCP7*, *TCP8*, *TCP14*, and *TCP15*, have been identified as promoters of flowering, while *TCP20*, *TCP22*, and *TCP23* act as inhibitors [14,15,20,31,32]. In addition, Class II TCPs have been found to be positive regulators of the flowering pathway, such as *TCP2* and CIN *TCPs* (*TCP3*, *TCP4*, *TCP5*, *TCP10*, *TCP13*, *TCP17*, *TCP24*) [17,18,19]. In soybeans, *GmTCP13* and *GmTCP15*, which belong to the CIN subclass of Class II, have been shown to enhance flowering [21,22]. Interestingly, to date, no Class I *GmTCP* gene has been identified as a regulator of flowering. Our study revealed that the Class I *TCP* gene *GmTCP40* is also a flowering activator, highlighting the need to investigate the potential roles of other Class I *GmTCPs* in photoperiodic regulation pathways.

In *Arabidopsis*, the flowering integrator *FT* and *TERMINAL FLOWER1* (*TFL1*) play antagonistic roles in regulating flowering with differences in their surface charges resulting from conserved amino acid variations [33,34]. Class I TCP members are potential candidates mediating the differential activities of FT and TFL1 [34]. In soybean, 10 FT homologs play antagonistic roles in regulating flowering with differences in conserved amino acids [35,36,37]. The functional divergence of these homologs may parallel that of FT and TFL1 and could be elucidated through Class I GmTCPs. In our study, we revealed that *GmTCP40*, a Class I *GmTCP*, promotes flowering. However, further exploration of the regulatory functions of other Class I *GmTCPs* is crucial for gaining a comprehensive understanding of photoperiodic regulation pathways in soybeans.

The *TCP* genes that promote flowering are regulated mainly by controlling downstream genes to regulate flowering. For example, *TCP7* promoted flowering by regulating the expression of *SOC1* and *FT* [14]. *TCP15* enhanced expression of *SOC1* [15]. *TCP5*, *TCP13*, and *TCP17* regulated the expression levels of both *CO*, *FT*, *AP1*, *FUL*, and *LFY* [19]. Similarly, *GmTCP40* induced the expression of *GmFT2a*, *GmFT5a*, *GmAP1s*, *GmSOC1a,* and *GmSOC1b* to promote flowering (Figure 3). Previous studies have demonstrated that *TCP* genes promote flowering by controlling downstream genes. TCP7 and TCP15 promote flowering by binding to the promoter of *SOC1* [14,15], and GmTCP15 binds to the promoter of *GmSOC1b*, a homolog of *SOC1* [22]. In *Arabidopsis*, TCP5, TCP13, and TCP17 induce flowering by binding to the promoter of *AP1* [19], a gene that plays an important role in floral organ development [38]. We found that similar to these *Arabidopsis* TCPs, GmTCP40 binds to the promoter region of *GmAP1a* and does not bind to the promoter region of *GmSOC1a* or *GmSOC1b* (Appendix A). Soybean has four *AP1* homologs, and the *gmap1* quadruple mutant shows delayed flowering, whereas overexpression of *GmAP1a* leads to early flowering [39]. GmAP1a directly binds to the promoter of *Dt1*, a repressor of flowering, to repress its expression [40]. Thus, GmTCP40 may promote flowering in soybean by promoting the expression of *GmAP1a* and potentially inhibiting the expression of *Dt1*. These findings contribute to our understanding of the photoperiodic flowering pathway.

In soybean, FDc interacts with GmFT5a to directly co-induce *GmAP1a* expression [40]. Soybean FD homologs, such as GmFDL12 and GmFDL19, interact with both GmFT2a and GmFT5a, while others, such as GmFDL06, interact exclusively with GmFT5a, and GmFDL15 interacts specifically with GmFT5b [41,42]. The interaction of GmTCP13 with GmFT2a and GmFT5a was verified by bimolecular fluorescence complementation assays [21]. In *Arabidopsis*, TCP5, TCP13, and TCP17 interact with the FT-FD complex to bind to the AP1 promoter [19]. This indicates that GmTCP40 may interact with the GmFT5a-FDc complex to regulate the expression of *GmAP1a*. This mechanism could play an important role in soybeans, especially in regulating flowering time. Further research on the interaction between GmTCP40 and the GmFT5a-FDc complex would help refine the soybean photoperiodic flowering regulatory network.

In a previous study of soybean, two nonsynonymous SNPs were identified in the CDS of *GmTCP13* [21]. In addition, a nonsynonymous CDS SNP and seven additional upstream SNPs were observed in the *TCP* homolog *BRC1* (*Glyma.06G210600*), which has been proposed as a candidate gene involved in regulating branch development [43]. In contrast, SNPs in *GmTCP40* were exclusively detected in the promoter region (Figure 5A). Therefore, the TCP protein is relatively conserved in soybeans without any instances of premature termination. A previous study showed that, compared with varieties carrying other haplotypes, *GmTCP13-*Hap3 varieties displayed a significantly delayed flowering [21]. Similarly, *GmTCP40*-Hap6 varieties also exhibited a later flowering time and were distributed mainly in South China (SC) (Figure 5B,C).

Deciphering the genetic mechanisms underlying soybean flowering time and regional adaptability is a crucial objective for breeders because of the extensive distribution of soybean varieties resulting from abundant natural variation and diverse combinations of genes and QTLs regulating flowering time. A natural variant of *GmELF3* provides soybean plants with an extended juvenile phase that enhances their adaptability in tropical regions [44,45]. In addition, natural variants of *GmPRR3a* and *GmPRR37*/*3b* contribute to soybean adaptation in high-latitude regions [46,47]. The variations in the promoter likely endow *GmTCP40*-Hap6 with a moderate but appropriate level of activity, leading to late flowering and adaption to low latitudes. Developing kompetitive allele specific PCR (KASP) markers based on these variations and utilizing the markers for genotyping could facilitate efforts to breed soybean varieties with the optimal flowering time for the target planting area, which may facilitate the introduction of soybeans varieties in China.

## Figures and Tables

**Figure 1 biomolecules-14-00465-f001:**
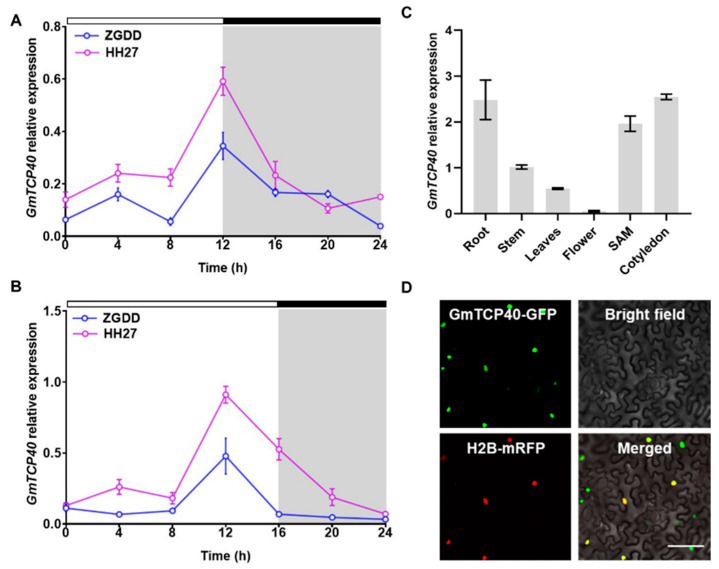
Expression pattern analysis of *GmTCP40*. (**A**,**B**) Expression levels of *GmTCP40* in the leaves of soybean varieties “Heihe27 (HH27)” and “Zigongdongdou (ZGDD)” under short-day (SD) (12 h light/12 h dark) (**A**) and long-day (LD) (16 h light/8 h dark) (**B**) conditions. The white and gray bars represent light and dark periods, respectively. (**C**) Expression levels of *GmTCP40* in different organs of the variety “Jack”. (**D**) Subcellular localization of the GmTCP40-GFP fusion protein in *Nicotiana benthamiana* leaves. Scale bar, 20 μm. The data are presented as the mean ± SD of three biological replicates.

**Figure 2 biomolecules-14-00465-f002:**
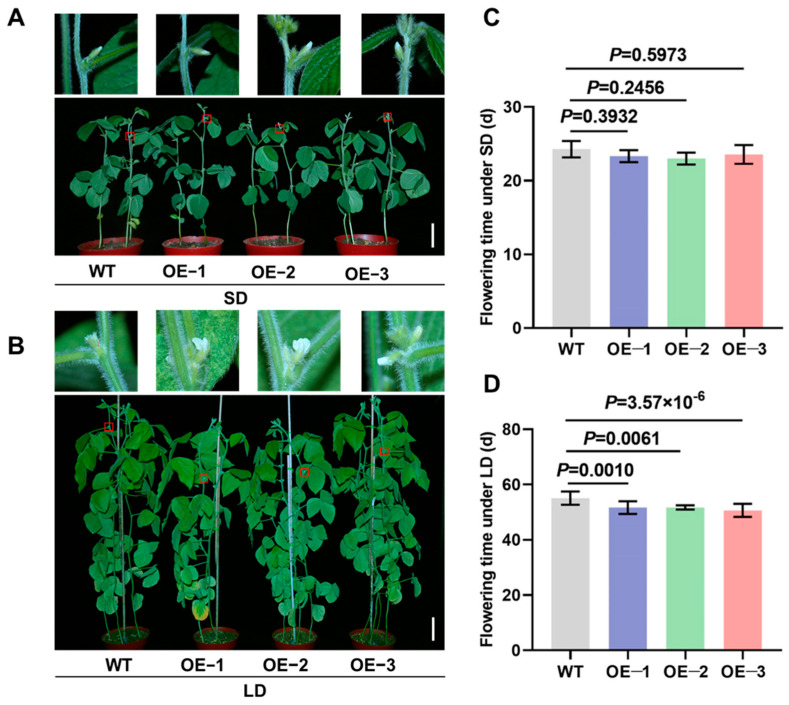
Overexpression of *GmTCP40* promotes flowering in soybean. (**A**,**B**) Flowering phenotypes of the three *GmTCP40*-overexpression lines (OE-1, OE-2, and OE-3) and wild-type (WT) plants under short-day (SD) (12 h light/12 h dark) (**A**) and long-day (LD) (16 h light/8 h dark) conditions (**B**). (**C**,**D**) Flowering times of OE-1, OE-2, and OE-3 lines and the WT plants under SD conditions (**C**) and LD conditions (**D**). Scale bar, 10 cm.

**Figure 3 biomolecules-14-00465-f003:**
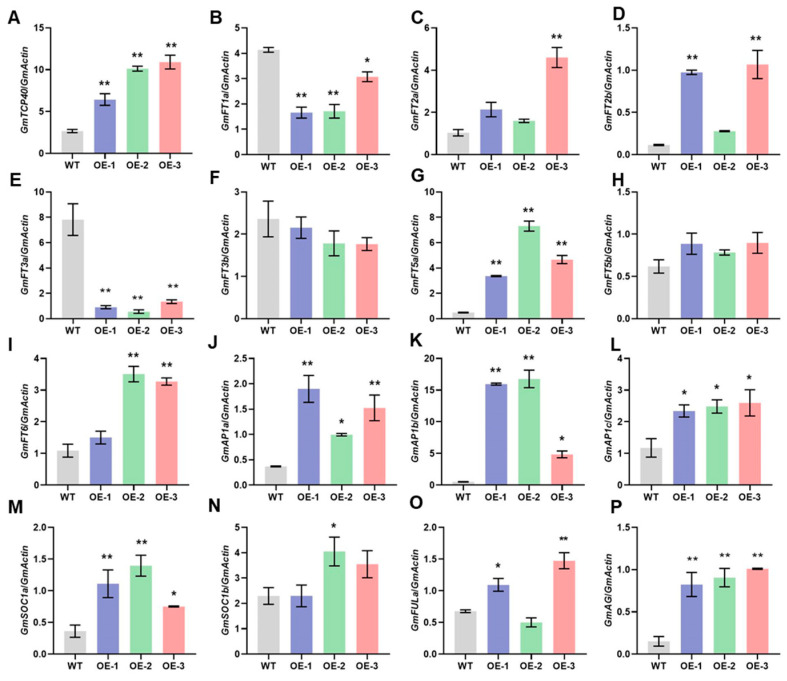
Expression levels of flowering-related genes in WT plants and *GmTCP40*-overexpressing transgenic lines under long-day conditions. (**A**–**P**) Leaves from three *GmTCP40*-overexpression lines (OE-1, OE-2, and OE-3) and wild-type (WT) plants under long-day (LD) (16 h light/8 h dark) conditions were harvested for qRT-PCR analysis of the expression levels of *GmTCP40* (**A**), *GmFT1a* (**B**), *GmFT2a* (**C**), *GmFT2b* (**D**), *GmFT3a* (**E**), *GmFT3b* (**F**), *GmFT5a* (**G**), *GmFT5b* (**H**), *GmFT6* (**I**), *GmAP1a* (**J**), *GmAP1b* (**K**), *GmAP1c* (**L**), *GmSOC1a* (**M**), *GmSOC1b* (**N**), *GmFULa* (**O**), and *GmAG* (**P**). The relative expression level was normalized to that of *GmActin*. The data are presented as the means ± SDs of three replicates (** *p* < 0.01; * *p* < 0.05).

**Figure 4 biomolecules-14-00465-f004:**
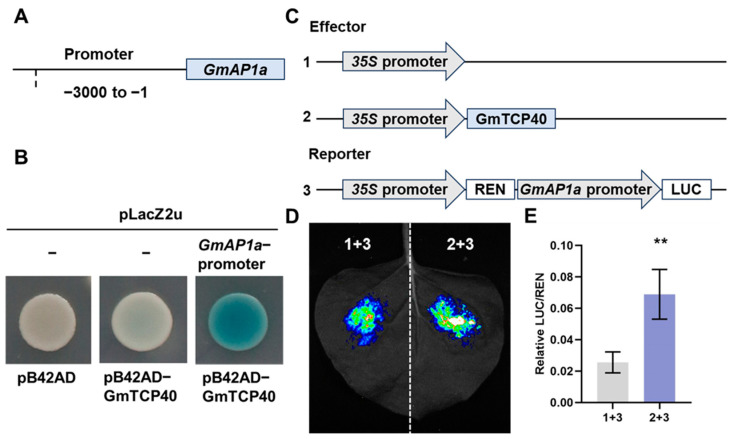
GmTCP40 activates transcription of the *GmAP1a*. (**A**) The 3 kb promoter of *GmAP1a* was subcloned into the pLacZ2u vector. (**B**) The interaction between GmTCP40 and the *GmAP1a* promoter was examined by yeast one-hybrid (Y1H) assays. The transformants were assessed on SD/-Trp/-Ura media supplemented with 20 mM X-gal, Galactose (Gal), and Raffinose (Raf). Empty vectors served as the negative controls. (**C**) *GmTCP40* was inserted into the effector construct pGreenII 62-SK, and the *GmAP1a* promoter was ligated into the reporter vector pGreenII 0800-LUC. Empty pGreenII 62-SK was used as the negative control. (**D**,**E**) The association of GmTCP40 with the *GmAP1a* promoter was investigated using transient luciferase reporter assays. The data are presented as the means ± SDs of three replicates (** *p* < 0.01).

**Figure 5 biomolecules-14-00465-f005:**
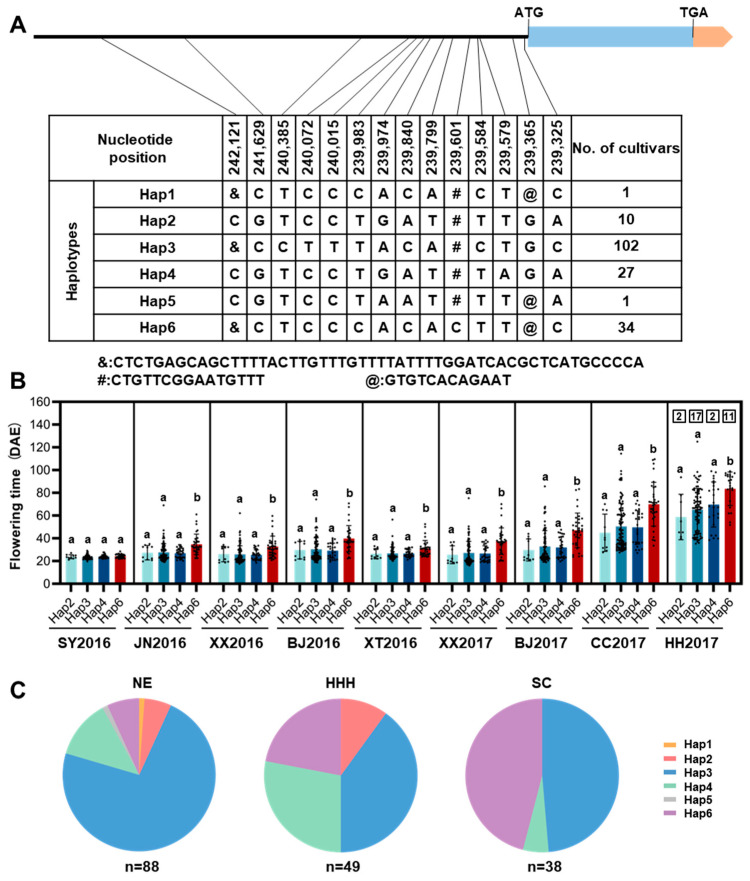
Analysis of the haplotypes of *GmTCP40* in 175 soybean varieties. (**A**) Natural variation in the nucleotide sequence of *GmTCP40*. Blue and orange represent the coding region and UTR, respectively. The black line represents the promoter. (**B**) Flowering time of soybean varieties harboring major *GmTCP40* haplotypes. Each dot represents a soybean variety and the number within each box indicates the number of varieties that did not flower. SY2016, JN2016, XX2016, BJ2016, XT2016, XX2017, BJ2017, CC2017, and HH2017: Sanya (18°18′ N, 112°39′ E) in 2016, Xiangtan (27°40′ N, 112°39′ E) in 2016, Jining (35°26′ N, 116°35′ E) in 2016, Xinxiang (35°08′ N, 113°45′ E) in 2016 and 2017, Beijing (40°13′ N, 116°33′ E) in 2016 and 2017, Changchun (43°50′ N, 124°82′ E) in 2017, and Heihe (50°24′ N, 127°49′ E) in 2017, respectively. DAE (days after emergence). The data are presented as the means ± SDs, and a and b indicate significant differences determined by Duncan’s test at *p* < 0.05. (**C**) Geographical distribution of soybean varieties harboring different alleles of *GmTCP40*. NE: Northeast China, HHH: Huang-Huai-Hai, SC: South China.

## Data Availability

Data are contained within the article and Appendix A.

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
