# Peer review of "GmTCP40 Promotes Soybean Flowering under Long-Day Conditions by Binding to the GmAP1a Promoter and Upregulating Its Expression"

_biomolecules, 2024, doi:10.3390/biom14040465_

Round 1
Reviewer 1 Report
Comments and Suggestions for Authors
Dear Authors,
I find your manuscript GmTCP40 promotes soybean flowering under long day conditions by binding to the GmAP1a promoter and upregulating its
expression interesting to read and believe that this findings will have influence on future breeding programmes. It is important for soybean, especially studies from China, where gene pool is the highest.
Introduction
could be extended with the importance of photoeriod in various time zones and explanation why did you choose to study this, why is this important to your location, what is the background looking from the several aspects of production and specific location (envirnments), not only from genetics pont of view.
Materials and methods
how many pots were sown, how many seeds, what was the emergence? which substrate was used?
I am wondering - what about other morphological parameters and their response to photoperiod?
What about the correlation of this gene and soybean yield?
Discussion
In disscussion there is too much enumeration. Please provide - why is your findings important in field growing of soybean, what are recomendations for breeding programs?
All the best
Comments on the Quality of English LanguageDear Editor,
thank you for invitation to Rewiev. This study seems interestign, but I put some main remarks.
All the best
Author Response
Dear,
Please see the attachment

Reviewer 2 Report
Comments and Suggestions for Authors
Please the authors to answer these questions to clarify the scientific work:
How did the authors choose to characterize GmTCP40 rather than another GmTCP? Did you select a random representative of Class 1?
Why did the authors use different soybean varieties to analyze the expression pattern of GmTCP40 and investigate its tissue-specific expression profiles?
Where were sequences downloaded for phylogenetic analyses?
Where was the GmTCP40 CDS sequence downloaded from?
Please reanalyze the data for Figure 2.D. The statistically significant difference is not clear!
Reviewer 3 Report
Comments and Suggestions for Authors
The work shows a complete study of the GmTCP40 gene and its involvement in the flowering time character. The most relevant results show that the lines with the overexpressed gene flower slightly later than the WT lines (about 4 days), and above all that one of the haplotypes found in a panel of 175 varieties, haplotype 6, with a deletion of 14 bp in CAAT-box elements could lead to functional differences in this gene. Haplotype 6 exhibited a later flowering time in the eight environments and its frequency decreased with increasing latitude in China. The authors also demonstrate that the GmTCP40 is capable of binding to the AP1a promoter and that the overexpressed lines also have greater expression of the AP1, SOC genes. FUL, Ag and different expression of 6 FT genes.
The work is well written but to be of greater interest I think some aspects could be improved. A pdf is attached with more specific questions, in general review the italics, there are already many genes and gene abbreviations that do not present them. The pdf also indicates that a couple of supplementary tables should be added, with the environments evaluated, indicating sowing date and flowering times (the reader does not know which environments are SD or LD), and another with the haplotypes of the 175 varieties and their flowering values.
Regarding the discussion, there is a lot of talk about the role of other TCPs, both in Arabidopsis and soybean, but the results obtained are hardly discussed. For example, more should be said about which FT genes have greater or lesser expression in OE lines, for example promoters such as FT2a and 5a have greater expression, while other repressors such as FT1a have lower expression.

Comments on the Quality of English LanguageEnglish is correct.
Reviewer 4 Report
Comments and Suggestions for Authors
Review of Biomolecules-2914325 manuscript
The manuscript describes experimentation to understand the action of the GmAP1a promoter and GmTCP40 protein. The manuscript describes creation of vectors to elucidate the molecular and function of the GmTCP40 protein including yeast single hybrid sytem to investigate promoter interaction. The manuscript describes the creation of a GFP fusion of the GmTCP40 protein to identify the subcellular compartmentalization of the protein. The manuscript was easy to read. However, there was a decided lack of information in the methods section. Most of my commentary comes from that section. Many of the procedures are poorly described and cannot be used to recreate the experiments. These must be improved.
The supplemental data has to have a diagram of the Vector pTF101-GmTCP40.
The vector needs to be deposited in an accessible repository or contact info must be included to get the plasmid DNA.
What was the source of the “Liberty Link” strips?
Describe how the “recombinant vector was introduced into Agrobacterium tumefaciens”.
How was the SNP calling done?
What program was used to map the reads to the reference?
What was the reference for the SNP calling?
What program was used to perform the “association analysis”?
How was the “fusion of Gm TCP40 to pB42AD vector” performed.
Need contact info for the pB42AD vector and a diagram of the GmTCP40-pB42AD construct.
How was the Gm AP1promoter regions subcloned into pLacZ2u vector? What kit was used?
List the sizes of promoter regions of GmSOC1a inserted into pLacZ2u. Need contact info for the vector. Need a diagram of the construct in supplementals.
How was the pLacZ2u vectors “transformed into the yeast strain EGY48” and why was that strain chosen? Where did you get the vector? (Clontech?)
Where did you get the SD media? What was the significance of -Trp/-Ura briefly.
How was GmTCP40 inserted into pGreenII 62-SK? Should have a diagram of the Y1H system used in the supplemental info.
How were the constructs introduced into Agrobacterium strain EHA105 and where did you get it? (pSoup and Zhuangmeng, China) is not descriptive enough to figure out where to get the reagents or how it was done.
What was the transformation procedure for the tobacco leaves?
What was the source of the Dual-Luciferase Reporter gene assay kit and how you used it.
Who determined there were 54 TCPs in soybean and did they do the classifications? Cite the source.
How was GmTCP40 made? What kit? Need a diagram of the construct in Supplementals.
How was the GmTCP40-GFP construct infiltrated into tobacco? Hopefully you have described this procedure earlier and can say “as previously described”.
Citation needed for this statement in the discussion section “However, only one class I GmTCP gene,GmTCP40, has been shown to promote flowering.”
Where did you get the MEGA 7 Software?
Round 2
Reviewer 4 Report
Comments and Suggestions for Authors
Re-review of Biomolecules-2914325
The methods section has been augmented as it should. This has met my concerns. I will note that a complete lab protocol is not needed, just enough information so that a skilled technician could reproduce the experiment. An option would be to put your lab protocols in a repository like Protocols.io and then you could refer to them using a single URL.
There was extensive revisions in the other sections which make the manuscript much more readable.
I will note that the legend of Supplementary Figure S2 needs to explain what panels A-P represent.
